# Clinical and Radiological Features of Interstitial Lung Diseases Associated with Polymyositis and Dermatomyositis

**DOI:** 10.3390/medicina58121757

**Published:** 2022-11-30

**Authors:** Stefano Palmucci, Alessia Di Mari, Giovanna Cancemi, Isabella Pennisi, Letizia Antonella Mauro, Gianluca Sambataro, Domenico Sambataro, Federica Galioto, Giulia Fazio, Agata Ferlito, Fabio Pino, Antonio Basile, Carlo Vancheri

**Affiliations:** 1Department of Medical Surgical Sciences and Advanced Technologies “GF Ingrassia”, University Hospital Policlinico “G. Rodolico-San Marco”, 95123 Catania, Italy; 2Radiology Unit 1, University Hospital Policlinico “G. Rodolico-San Marco”, 95123 Catania, Italy; 3Regional Referral Center for Rare Lung Disease, Department of Clinical and Experimental Medicine, University Hospital Policlinico “G. Rodolico-San Marco”, 95123 Catania, Italy; 4Artroreuma S.R.L., Rheumatology Outpatient Clinic Associated with the National Health System, 95030 Mascalucia, Italy

**Keywords:** polymyositis, dermatomyositis, lung disease interstitial, multidetector computed tomography, autoimmune diseases

## Abstract

Polymyositis and dermatomyositis are autoimmune idiopathic systemic inflammatory diseases, characterized by various degrees of muscle inflammation and typical cutaneous lesions—the latter found in dermatomyositis. The underlying pathogenesis is characterized by a high level of uncertainty, and recent studies suggest diseases may have different immunopathological mechanisms. In polymyositis, components of the cellular immune system are involved, whereas in dermatomyositis, the pathogenesis is mainly mediated by the humoral immune response. The interstitial lung disease occurs in one-third of polymyositis and dermatomyositis patients associated with worse outcomes, showing an estimated excess mortality rate of around 40%. Lung involvement may also appear, such as a complication of muscle weakness, mainly represented by aspiration pneumonia or respiratory insufficiency. The clinical picture is characterized, in most cases, by progressive dyspnea and non-productive cough. In some cases, hemoptysis and chest pain are found. Onset can be acute, sub-acute, or chronic. Pulmonary involvement could be assessed by High Resolution Computed Tomography (HRCT), which may identify early manifestations of diseases. Moreover, Computed Tomography (CT) appearances can be highly variable depending on the positivity of myositis-specific autoantibodies. The most common pathological patterns include fibrotic and cellular nonspecific interstitial pneumonia or organizing pneumonia; major findings observed on HRCT images are represented by consolidations, ground-glass opacities, and reticulations. Other findings include honeycombing, subpleural bands, and traction bronchiectasis. In patients having Anti-ARS Abs, HRCT features may develop with consolidations, ground glass opacities (GGOs), and reticular opacities in the peripheral portions; nonspecific interstitial pneumonia or nonspecific interstitial pneumonia mixed with organizing pneumonia have been reported as the most frequently encountered patterns. In patients with anti-MDA5 Abs, mixed or unclassifiable patterns are frequently observed at imaging. HRCT is a sensitive method that allows one not only to identify disease, but also to monitor the effectiveness of treatment and detect disease progression and/or complications; however, radiological findings are not specific. Therefore, aim of this pictorial essay is to describe clinical and radiological features of interstitial lung diseases associated with polymyositis and dermatomyositis, emphasizing the concept that gold standard for diagnosis and classification–should be based on a multidisciplinary approach.

## 1. Introduction

Polymyositis (PM), dermatomyositis (DM), and anti-synthetase syndrome (AS) are conditions included among idiopathic inflammatory myopathies (IIMs), characterized by various degrees of muscle inflammation, with common involvement of joints, lung, and skin—the latter mainly represented by typical cutaneous lesions [1]. They are rare disorders, with a prevalence of about 14–17 persons per 100,000 [2]. 

Respiratory disease is a typical manifestation in these conditions, and the frequency and severity depend on the underlying inflammatory myopathy subtype [3]. The relationship between idiopathic inflammatory myopathy and diffuse Interstitial Lung Disease (IIM-ILD) was described for the first time in 1956 by Mills and Matthews [4]. ILD occurs in 9.19–60% of PM/DM patients associated with worse outcomes, showing an estimated excess mortality rate of around 40% [5,6]. Lung involvement may also appear, such as a complication of muscle weakness, mainly represented by aspiration pneumonia or respiratory insufficiency [7].

Patients with CTD should be evaluated using high-resolution computed tomography (HRCT) to investigate the presence of ILD, which may be the first manifestation. Chest X-ray can identify parenchymal abnormalities, but they only appear in advanced stage [8].

HRCT is the gold standard for early diagnosis and identification of ILD patterns: it plays an important role in the management of these diseases in order to start medical treatment and to monitor disease progression and/or complications.

This article focuses on the available data regarding clinical features, prevalence, laboratory findings, radiological features, predictors, and prognosis of ILD-CTD. Furthermore, we attempt to increase awareness on the role of HRCT in the characterization of radiological patterns of ILD, which are also related to the antibody positivity (briefly summarized in Table 1).

## 2. Methods

Literature research was carried out through the Pubmed databases using the following queries “myositis AND interstitial lung disease”.

Literature retrieval started on September 2022 and was concluded in November 2022 including articles published in the last 5 years from 2017 to 2022.

Through title and abstract review, 720 studies were further excluded for the following reasons: non full-text article, case reports or series with fewer than 10 patients (*n =* 697), and recurrent articles from the same authors (*n =* 3). Therefore, 127 articles were subjected a full-text review, and after that, 20 articles were excluded because they were not in English (*n =* 3) and not relevant (*n =* 18). Finally, 107 publications relevant to the purpose were selected for this review. 

The images were obtained by retrospective file analysis from our Radiology Information System (RIS) and Picture Archiving and Communication System (PACS) for all consecutive patients with PM and DM confirmed diagnosis who perform HRCT evaluation between March 2017 and October 2022 (Figure 1).

## 3. Clinical Features

Both PM and DM may exhibit a varying degree of muscle weakness, usually developing slowly over weeks to month; in some cases, acute or subacute onset of weakness may be observed [9].

The muscular weakness is relatively symmetric, with predominantly proximal location. Diseases are not associated with sensory loss or ptosis with sparing of extraocular muscles, which are typically observed in other muscular disorders (myasthenia) [4,10]. In the late stages of PM and DM, distal muscle weakness, which affects fine motor movements, can occur. 

Respiratory failure, dysphagia, nasal speech, and aspiration pneumonia may occur in late phases, due to involvement of diaphragm and thoracic muscles or pharyngeal muscles [9]. 

The clinical presentation of ILD is variable, spreading from asymptomatic to exertional dyspnea, exercise intolerance, and non-productive cough; bibasal dry crackles may be heard on chest auscultation, with reduced chest movement due to respiratory muscle weakness [11,12]. CADM, DM, and AS—in descending order by frequency—are often associated with skin involvement. As reported by Yang SH et al. “some presenting symptoms include myalgias, muscle weakness, and a combination of ILD, Raynaud’s phenomenon, seronegative arthritis of the distal joints, fever, mechanic’s hands, and a skin rash that is different from the heliotrope erythema” [9].

The most specific manifestations are as follows: (i) “Mechanic’s Hands” and “Hicker’s Feet” (Figure 2)—a hyperkeratotic and non-pruritic eruption with scaling, fissuring, and hyperpigmentation, respectively, of the ulnar surface of the first finger and the radial surface of the second and third fingers of the heels; (ii) Gottron’s rash and papules (Figure 3), which consist of violaceous macules or raised papules on the bone prominence, but typically located in the little joints of the hands [13]. These signs are almost pathognomonic for IIMs. Other typical cutaneous manifestations are Heliotrope rash (a violaceous erythema of the periorbital and nasolabial surface) and a red macular or papular rash localized in arms, legs, or chest (respectively named Shawl Sign, Holster Sign, and V Sign) [14]. 

Inflammatory Arthritis is very common in IIMs, and part of the classic triad for the classification of AS (myositis, ILD, arthritis). It can be asymmetric and oligoarticular (mainly with an onset during the disease follow-up) or resembling Rheumatoid Arthritis, with possible erosions (mainly at the diagnosis) [15].

Raynaud’s phenomenon (RP) (Figure 4) can be present in about 30–40% of patients [16]. RP is classically associated with the presence of a disease of the Scleroderma Spectrum Disorders (SSDs), and it could not be excluded that some IIMs could have another condition (among SSDs) in overlap. The study of RP-IIMs with Nailfold Videocapillaroscopy (NVC) can highlight a scleroderma pattern; however, significant angiogenesis can often be found [17]. NVC can be useful to support the diagnosis of IIMs even without the presence of RP [13,18].

Digital clubbing and hypertrophic osteoarthropathy are rare extrapulmonary manifestations of PM-DM ILD [19,20]. Several studies have focused on the possible prognostic value of digital clubbing in patients with ILD, reporting discordant opinions [19,21]. Essaouma et al. have found a prevalence of 31.1% of digital clubbing and hypertrophic osteopathy in adults with ILD [22]. According to the paper by Shiraishi K et al., digital clubbing has been associated with high serum Krebs von den Lugen (KL-6) levels and decreased pulmonary function; however, it has been found with heterogeneous frequency in ILDs, with percentage values of 20.8% in non-CTD-ILD cases, 12% in CTD-ILD patterns, and 25% in ILD related to Scleroderma. No digital clubbing has been found among patients with DM-ILD [19].

Finally, constitutional symptoms, such as unexplained fever, myalgia, fatigue, and weight loss, can be already present in up to 20% of patients [17].

### 3.1. Laboratory Findings

As already reported, an increased serum level of muscular enzymes can be useful to suspect the disease; however, serum muscular enzymes could be not correlated with the clinical conditions referred by the patients [23]. A great improvement in diagnostic accuracy could be obtained through the identification of specifics autoantibodies for these diseases [23]. Antinuclear antibodies (ANA) can be found, generally with a nucleolar or cytoplasmic pattern; however, a seronegativity for ANA cannot exclude the possible presence of anti-synthetase antibodies [23]. Classically, these autoantibodies are divided in Myositis-Specific and Myositis-Associated Antibodies (MSAs and MAAs, respectively). Among MSAs, the most common is Jo1, associated with the classic triad of AS: myositis, arthritis, and ILD. Other rare anti-synthetase antibodies are PL7, PL12, EJ, OJ, KS, Zo, and Ha [24]. The latter group is generally associated with a prevalent lung involvement [24].

MAAs included a group of autoantibodies also present in other Connective Tissue Diseases (CTDs) such as Anti-Ro52, Pm/Scl, U1-RNP, and anti-Ku [24]. Although the specificity for IIM is lower, they can be useful to suspect an autoimmune underlying condition, and they are associated with the presence of myositis and ILD.

### 3.2. Classification Criteria

The first classification for PM/DM was produced by Bohan and Peter in 1975, and it is still the most widely used despite several limitations [25,26]. A new approach was proposed in 1990, aimed to classify IIMs according to the autoantibody positivity [27]. A recent, joint effort of the American College of Rheumatology and European League Against Rheumatism (ACR and EULAR, respectively) proposed and validated a new core of classification criteria for IIMs with good performances [28]. The proposal and validation of AS criteria by ACR and EULAR is currently ongoing.

### 3.3. Pulmonary Involvement in IIMs

The first description of pulmonary involvement was reported by Mills and Mathews in 1956 [4]: these authors described interstitial lesions’ development in DM. Lung involvement occurs as a serious complication of these diseases; in DM-PM, it represents the most common site of extra-muscular location [29]. 

Several papers have reported epidemiological data regarding ILD in DM-PM: according to some authors, it may develop in 23.1–65% of PM/DM patients [30,31,32]. In other articles, ILD prevalence in DM/PM patients ranges from 19.9% up to 78%; this wide interval of disease prevalence is dependent on the method used for its detection [33]. ILD may precede the diagnosis of DM/PM in 13 up to 37.5% of patients, and it is generally associated with fever and joint involvement [33]; in other cases, it may be depicted contemporary to the muscle involvement. The ILD development represents a factor of increased morbidity and mortality in PM/DM patients when it is revealed at admission [30,34]. 

It is also well known that ILD may be the first or the only manifestation of CTDs, and up to 69% of patients may have ILD as a unique manifestation of disease [29,35]. Lung involvement may be also depicted as response to treatment in these diseases, resembling the appearance of a drug-induced lung disease [29].

## 4. Radiological Features

Chest X-ray may demonstrate imaging features of ILDs, even if disease findings are not specific and generally detected in advanced stages. A restrictive imaging pattern may be suspected in patients having large number of reticulations and honeycombing areas. HRTC is the modality of choice for assessing the ILD pattern: it is an imaging technique that allows one to get the diagnosis in early phases, to confirm the location and extent of disease, and to estimate its activity. 

Thanks to the improvement of technology, various diagnostic methods have been developed with the aim of reducing the radiation dose. In 2021, the first clinical photon CT (PCCT) was introduced; this CT technique has many advantages such as spectral imaging capability, higher spatial resolution, minimization of electronic noise, and reduction of radiation exposure. In literature, some studies have compared HRCT and PCCT: the latter seems to reproduce better results, even if further studies are needed to assess the diagnostic impact of PCCT having improved image quality [36].

The impact of HRCT on prognosis has been investigated in CTD patients with associated pulmonary manifestations consisting of ILD, ventilation defects, and aspiration pneumonia.

Regarding ILD patterns, in a previous paper, Cottin [37] investigated interstitial pneumonia associated with IIMs, highlighting the relationship between the onset of lung involvement and the muscular and/or cutaneous manifestations. This study included 17 patients (4 males, 13 females, aged 51.7 ± 10.8 years) selected according to the following criteria: (1) diagnosis of DM/PM according to criteria of Bohan and Peter, (2) presence of diffuse infiltrative lung disease on chest CT, and (3) surgical lung biopsy.

Ten patients had a diagnosis of DM, four of PM, and three of ADM; in the latter case, the diagnosis was established in the presence of a skin rash characteristic of DM/PM, associated with increased levels of muscle enzyme. 

In 24% of cases, the lung involvement preceded the cutaneous and muscular manifestations. The main HRCT abnormalities observed for each patient were as follows: consolidations (6 out of 17 cases), ground-glass attenuation areas (5 out 17 cases), reticular opacities (5 out of 17 cases), and peribronchovascular thickening (in only one patient). Air-space opacities or consolidations were also reported, with a small size and showing a diffuse and/or peribronchovascular distribution; in only one patient, there was a postero-basal large consolidation with a peripheral location [37]. Histologically, as reported in 2003 [37], non-specific interstitial pneumonia (NSIP) is the most commonly encountered pattern (11 out of 17 cases); more in detail, in the series reported by Cottin, cellular NSIP was observed in two cases, a mixed inflammatory and fibrotic NSIP was found in five subjects, and in the remaining patients a predominantly fibrotic NSIP pattern was found [37]. The remaining diagnoses included two cases of usual interstitial pneumonia (UIP), two cases of organizing pneumonia (OP), and one case of lymphoid organizing pneumonia. In only one case, the pathological diagnosis NSIP was not confirmed at imaging, since that HRCT features were more suggestive of OP. Therefore, as widely reported in literature, NSIP, OP, and UIP are the three radiological patterns described in ILD related to myositis-dermatomyositis.

The NSIP pattern is mainly represented by two main subtypes: fibrotic type and cellular type, the latter with a better prognosis due to a successful treatment response. On HRCT, the most common findings are ground-glass and fine reticulations (Figure 5 and Figure 6), predominantly in the lower lung zone with subpleural sparing. Generally, these abnormalities have symmetrical and bilateral localization with an apicobasal gradient. 

Other manifestations include traction bronchiectasis (Figure 6 and Figure 7) and thickening of bronchovascular bundles—mostly associated with fibrotic NSIP or UIP/probable UIP— as well as lung volume loss, especially at the lower lobes [38]. In advanced disease, traction bronchiectasis, consolidations, and areas of microcystic honeycombing are often recognized. In some NSIP cases, HRCT may demonstrate peripheral areas of consolidations, also with perilobular appearance; in these cases, a mixed NSIP-OP pattern (Figure 8) may be revealed at imaging [39].

The main features in OP pattern are ground-glass opacities and consolidations (Figure 9), followed by other minor findings such as reticulations, bronchiectasis, interstitial nodules, interlobular septal thickening, reversed halo sign, airspace nodules, and halo sign; lesions have a generally subpleural/peribronchovascular and perilobular distribution that is predominantly bilateral and interesting the middle and the lower thirds of the lungs (Figure 10). OP can also manifest an overlap with idiopathic pulmonary fibrosis and NSIP. In this case, there are relatively less depicted ground-glass opacities and consolidations, and the lung is predominantly affected by reticular opacities with architectural distortion [40].

In the case series reported by Kang, 29 patients showed interstitial lung disease on HRCT scans; in 11 out of these 29 cases, the radiological pattern was UIP. In 13 cases, a cryptogenic organizing pneumonia (COP) pattern was depicted on HRCT scans, with peripheral consolidations located in basal regions, with or without ground-glass areas in a geographic distribution [30]. The NSIP pattern was found in only four cases, with ground-glass opacifications and bronchiectasis. In the remaining case, a combined appearance was found, due to the coexistence of UIP, COP, and pneumomediastinum [30]. HRCT features frequently seen in UIP include honeycombing, traction bronchiectasis, and bronchiectasis (Figure 11), which may be seen with the concurrent presence of ground-glass opacification and fine reticulation. Honeycombing is a distinguishing feature of UIP, and it must be present for a definite diagnosis of UIP: it refers to cystic airspaces of 3–10 mm diameter with thick, well-defined walls. It often presents as multiple layers of subpleural cysts on top of each other, but it may also present as a single layer. The typical distribution of UIP is subpleural with basal predominance, although some upper lobe may be involved [41].

### 4.1. Radiological Features of ILD Presentation/Progression in DM/PM

Many forms of IIM-ILD have slowly progressive or subclinical courses, whereas others are associated with rapidly progressing pulmonary involvement that results in worse outcomes [29,42]. 

The presentation and progression of IIM-ILDs generally follow one of these patterns:▪Asymptomatic/occult ILD (minimal or absent changes seen on HRCT) [43];▪Acute or rapidly progressive ILD (acute interstitial pneumonia and radiological evidence of acute respiratory distress syndrome (ARDS) due to diffuse alveolar damage) [44];▪Subacute or chronic ILD, having consolidations or ground-glass opacities on HRCT scans, and reproducing OP, NSIP, and OP/NSIP patterns [45,46,47];▪ Chronic progressive fibrosing ILD (reticulation and honeycombing with minimal GGOs on HRCT scans, related to fibrotic NSIP or UIP) [43];

Regarding subacute and chronic groups, as the disease progresses, consolidations typically resolve, and honeycombing and fibrosis become more common [10].

The impact of HRCT on prognosis has been investigated in PM/DM patients with associated ILD, and many observational and interventional studies have been conducted.

More in detail, in a cohort of 497 patients with PM/DM –ILD enrolled in a multicenter retrospective analysis of 44 Japanese institutions, the 3-year survival rate was 80%, with ILD being the major cause of death [48]. According to the myositis-specific autoantibodies associated with ILD, the one-year survival rate was 66% for anti-MDA5 antibody-positive patients, 96% for anti-ARS antibody-positive patients, and 95% for both antibody-negative cases [48].

Although rapid diffuse alveolar damage and pulmonary deterioration can occur in patients regardless of the initial HRCT pattern, Bonneffoy et al. highlighted that unfavorable evolution is constant when honeycombing is recognized on initial CT scans: in these cases, annual follow-up is recommended to assess disease progression [49].

Tanizaka et al. investigated the possible prognostic role of HRCT in patients with PM /DM, concluding that higher short-term mortality is associated with lower consolidation/GGOs patterns. In their study, which included 51 PM/DM-ILD patients, they highlighted how disease extent and severity at the time of diagnosis cannot explain mortality. Rather, the lack of response to initial treatment may be the culprit of short-term mortality. In fact, all deaths within 90 days were caused by progressive ILD, even if PM/DM diseases were not widely distributed at the diagnosis [50].

Furthermore, HRCT findings of honeycombing/traction bronchiectasis (Figure 11) or a histopathologic diagnosis of UIP are related to a poorer prognosis, although several studies reported a better overall survival in patients with myositis-associated ILD (including UIP) compared to idiopathic pulmonary fibrosis (IPF) [51].

According to a study including three different groups of patients, namely those with CTD-ILD, undifferentiated CTD (UCTD)-ILD, and IPF, the CT findings were not significantly different among three groups [52].

Another study asserted that in patients with CTD-ILD and UIP pattern on CT, straight edge sign, exuberant honeycombing sign, and anterior upper lobe sign are significantly more common than the signs in patients with IPF and UIP pattern [53].

#### Acute Type

The acute interstitial pneumonia is more common in DM than in PM; it is related to the development of diffuse alveolar damage. Pulmonary manifestations consist of diffuse and multiple ground-glass opacities with consolidations; these latter CT abnormalities are located around the broncovascular bundles of lower lobes. Moreover, the increasing attenuation due to edematous changes may reproduce a mosaic pattern; the edema is also associated with interlobular/intralobular interstitial thickening.

Although rapid diffuse alveolar damage and pulmonary deterioration can occur in patients regardless of the initial HRCT pattern, Bonneffoy et al. highlighted that unfavorable evolution is constant when honeycombing is recognized on initial CT scans: in these cases, annual follow-up is recommended to assess disease progression [49,54].

### 4.2. Relationship between Myositis-Specific Antibodies and ILD 

The autoantibody profile may be related with different IIM sub-phenotypes and aid in risk stratification, namely for Rapid Progressive-ILD (RP-ILD) development. 

The most important IIM sub-phenotypes include those associated with anti-synthetase antibodies (anti ARS-Abs) and DM associated with anti-melanoma differentiation-associated gene 5 antibodies (anti-MDA5).

Anti-synthetase antibodies are the most common autoantibodies observed in patients with either DM or PM, with an average prevalence of 20 and 29%, respectively [55]. Eight types of anti-synthetase antibodies directed against the aminoacyl-tRNA synthetase enzyme (anti ARS-Abs) have been identified, including the well-known anti-Jo-1 antibody and anti-PL7, anti-EJ, anti-OJ, anti-PL12, and anti-KS antibodies. However, anti ARS-Abs positivity is not synonymous with myositis and may also be seen in other collagen diseases and idiopathic interstitial pneumonia [56].

Although the clinical features are variables, being related to the different antibody profile, the ILF frequency is generally high (more than 90%). Other clinical symptoms, such as polyarthritis, Raynaud’s phenomenon, and mechanical hands, are also frequently detected. It is named as anti-synthetase syndromes (ASS), independent of myositis [54].

Anti-Jo1 is the most common anti ARS-Ab, being revealed in up to 60% among all ARS-Abs [57]. It has been associated with an increased rate of arthralgias [55,57], mechanics hands [55], and myositis. Although ILD could not be present at baseline, it ultimately develops in 84% of all patients, including the majority of those initially presenting with isolated arthritis as the only defining feature of the anti-synthetase syndrome [58].

Anti-PL-12 has been associated with the development of Raynaud’s phenomenon and isolated ILD [59]. Anti-PL-7 is associated with the development of a heliotrope rash, myositis, ILD preceding a diagnosis of myositis [59], and pericardial effusion. Both anti-PL-12 and anti-PL-7 have been associated with more frequent and severe lung involvement when compared with anti-Jo1 [58].

Pulmonary involvement in ASS has a subacute to chronic course and is often more responsive to initial therapy than in MDA5 Ab-positive cases.

#### 4.2.1. HRCT Findings of Patients with Anti-ARS Abs

In a study by Waseda et al., which included 67 patients with ILD presenting ASS, the most common HRCT pattern was the NSIP pattern (35 patients) followed by Fibrosing OP (FOP) pattern [56]. The FOP pattern is considered an intermediate pathology between NSIP and OP [54]. On HRCT scans, the most common encountered lesions were GGOs (98.4%), followed by reticular opacities (67.2%) and consolidations (48.4%); the distribution of lesions was homogeneous, mainly represented in the lower lung lobes (98.4%), around the bronchovascular bundles (73.4%), and in the periphery (95.3%) (Figure 12); it may be also accompanied by a loss of volume [56]. Hozumi et al. confirmed these findings, reporting that all CT findings—in patients positive for anti-ARS Abs—were associated with NSIP pattern or NSIP with OP pattern [46]. Similar findings were also found in other series [59]. 

Anti-melanoma differentiation-associated gene 5 (Anti-MDA-5) autoantibodies were initially known as anti-CADM-140 antibodies, due to their presence in patients with Clinically Amyopathic DermatoMyositis (CADM), who were clinically asymptomatic for myositis [54]. The presence of MDA-5 antibodies occurs almost exclusively in DM patients. These patients usually present with a rapidly progressive acute to subacute course of disease, often being admitted to emergency departments with severe dyspnea [54]; in fact, this autoimmune profile confers a higher risk of severe and progressive ILD and death [58]. To recognize the disease at the early stages may be lifesaving, so it is recommended testing for this antibody in all patients with underlying myositis or ILD [58]. 

#### 4.2.2. HRCT Findings of Patients with Anti-MDA-5 Abs

HRCT data of anti-MDA-5 patients have been scarcely reported due to their recent investigation [54]. Anti-MDA5 Ab positivity may be represented by an acute lung injury with severe OP pattern, leading to diffuse alveolar damage (DAD) without chronic lesions [55]. Tanizawa et al. [50] evaluated the CT images of 51 patients with PM/DM-ILD at the time of initial diagnosis: they found that those with consolidations or GGOs in the peripheral lower lung fields or along the bronchovascular bundles were more likely to be anti-MDA-5 Ab-positive. They also demonstrated that lower consolidation/GGOs pattern is a significant determinant of short-term prognosis not only in PM/DM-ILD patients, but also in subjects with anti-MDA-5 status. Hozumi et al. found that the most encountered HRCT pattern in MDA-5-positive DM patients was labeled as “unclassifiable,” consisting of a mixture of consolidations and GGOs associated with reticulations; this pattern was not typical of pure NSIP, OP, or UIP [46,60]. In other studies, most common ILD pattern found in anti-MDA 5 Abs IIMs was OP (up to 50%), followed by NSIP-OP pattern (30%) and NSIP (20%) [61].

Most recently, a striking similarity has been recognized between severe COVID-19 and anti-MDA5 DM [62]. Mehta et al. highlighted the similarity between the diseases with possible similar involvement of the lung, skin rashes, fever, fatigue, and myalgia; moreover, radiological features of COVID-19 pneumonia are comparable to those of ILD in anti-MDA5 DM, with presence of diffuse GGOs and peribronchovascular consolidations [62].

Common imaging findings based on the antibodies are shown in Table 2.

## 5. Pulmonary Comorbidities of ILD in Patients with PM/DM

Based on many reports published in literature, acute respiratory failure (ARF) is the most common cause of death in in patients with PM/DM [63]. Comorbidities, such as asthma, chronic obstructive pulmonary disease (COPD), infection/aspiration pneumonia, coronary artery disease (CAD), hypertension, and heart failure, have been related to PM/DM [63]. The etiology of ARF in these patients is still not well explained: the main theories include cardiac origin of diseases, related to hypertension and heart failure, and pulmonary origin diseases due to hypoventilation and pneumonia. Therefore, the primary pulmonary complication of PM/DM (pneumonia) and the primary cardiac complication (hypertension-related heart failure) contribute to the development of ARF and death. These findings imply that PM/DM itself is a risk factor for ARF, even in the absence of comorbidities [58]. 

In anti-MDA5-positive DM patients, there has been reported an increased occurrence of spontaneous pneumomediastinum (PNM), also known as mediastinal emphysema and Hamman’s Syndrome. Spontaneous PNM is caused by non-traumatic and ambiguous pathophysiological mechanisms, such as the underlying RP-ILD or vasculopathy, with an increased mortality of approximately 60% [64].

### 5.1. Pulmonary Ipertension 

Pulmonary hypertension (PH) is another possible manifestation of lung involvement, likely under-recognized, in patients with PM and DM. PH was first described in 1956; it has been reported with a prevalence between 7.9 and 16.39% in IIMs, which is less prevalent than percentages reported in other CTDs [65].

The exact etiopathogenesis if PH development has not yet been clarified: it could be consequential to hypoxemic vasoconstriction due to the progressive pulmonary fibrosis (so correlated with ILD). Another possible explanation retains PH secondary to a vessels remodeling, as happens in primary form of hypertension [43]: this theory is supported by high levels of pulmonary pressure and by the presence of other signs and symptoms related to prevascular pulmonary hypertension–such as the Raynaud’s phenomenon [65,66,67]. A French retrospective study revealed the possible association between PH and IIMs without extensive ILD, highlighting the worst outcome of these patients [68]. Therefore, further research is needed to provide greater clarity on the pathogenesis of PH.

Gasparotto et al. suggested an independent contribution of PH on the prognosis and survival rate of ASS [69]; for this reason, it is desirable to conduct a screening by echocardiography, and right heart catheterization is required for a definitive diagnosis of PH, namely for patients with possibility of treatment [70].

### 5.2. Pneumonia Secondary to Pulmonary Muscular Weakness

In patients with PM and DM, infectious pulmonary complications have been reported in a small series of PM/DM patients [31,71], representing an independent risk factor for early mortality [4,72]. Several factors may be implicated in this increased frequency of infections in PM/DM patients. Respiratory failure, related to the involvement of respiratory striated muscle (diaphragmatic, intercostal, and accessory muscle), can lead to hypoventilation, reducing the lung clearance. Other possible factors include the use of a high dose of glucocorticoids, steroid-sparing immunosuppressive treatment, and upper esophageal involvement [29,73].

Pyogenic infections are usually secondary to aspiration pneumonia that may be caused by nasopharyngeal dysfunction [31]. Marie and colleagues [31] studied 279 patients with inflammatory myopathy and found that two-thirds of pyogenic infections were related to aspiration pneumonia (Figure 13). One-fourth of these patients also had ILD on their chest imaging, and 17% of these patients died because of pneumonia-related complications within 1 year of PM/DM diagnosis. Several studies have shown that patients with ILD have a higher prevalence of abnormal acid exposure in the proximal and distal esophagus than those without ILD. The cause has not been clarified [74]; although it is standard for proton pump inhibitor therapy to be administered to help prevent microaspiration, further studies are needed to demonstrate long-term benefit. Opportunistic lung infections are most commonly caused by fungi, which accounted for about 40% of cases in a study conducted by Marie et colleagues in 2011 [31]. Pneumocystis Jirovecii and Candida albicans are responsible for the majority of opportunistic infections in DM/PM. To date, there are no clear and updated guidelines regarding prophylaxis in patients treated for IIM.

### 5.3. Drug-Induced Pneumonitis

Immunosuppression is the rationale for the treatment of IIM and IIM-ILD, and several immunosuppressive therapies have been proposed. In the acute presentation of ILD, steroids have been used as the first-line therapy. Cytotoxic agents, such as methotrexate, cyclophosphamide, and biologic therapies, have been largely proposed in the advanced stages of disease [29]. 

Some of these drugs can cause interstitial pneumonia, so that it could be difficult to distinguish these induced patterns of lung damage from pre-existing ILDs: their use remains controversial, and should be discussed among specialists, in order to evaluate the risk-benefit ratio of this kind of treatment [51]. 

Methotrexate and cyclophosphamide are included among the drugs that may cause more frequently interstitial pneumonitis. Most often methotrexate-induced pneumonitis can be found during the first year of therapy, and usually resolves with discontinuation of treatment; in addition, it is not related to the administered dose. In 2000, Imokawa et al. reviewed 123 cases reported in literature since 1969: typical chest radiographic findings included interstitial and mixed interstitial and alveolar infiltrates, with a predilection for the lower lung fields [75]. 

The main radiological patterns of methotrexate-related lung changes are represented by fibrotic disease with bronchiolitis obliterans and organizing pneumonia (previous termed as BOOP), or NSIP—the latter presenting with diffuse ground-glass opacities in early stage and basal fibrosis in advanced stages [76,77].

Cyclophosphamide can also cause lung toxicity with interstitial pneumonia that can occur in two different patterns: in an early phase of treatment (acute form) or after prolonged therapy (chronic form). Early-onset pneumonitis responds to discontinuation of the drug, whereas patients with late-onset pneumonitis may develop progressive pulmonary fibrosis associated with bilateral pleural thickening and show no response to cessation of treatment (Figure 14) [78].

Many of the newer antimetabolites, such as leflunomide, can cause pneumonitis–either if administered as monotherapy, or in combination with methotrexate [79]. Radiological features of patients treated with leflunomide are represented by ground-glass opacifications superimposed on pre-existing ILD (due to underlying disorder or induced by methotrexate; these features could have poor prognostic value). For this reason, leflunomide should not be used in patients with previous methotrexate-induced pneumonia and should be used with caution in patients with ILD [80].

Lung disease is commonly encountered also as a consequence of using the newer biologic agents—such as rituximab. The immunopathogenesis of newer biologic associated ILD is unknown. In contrast, several studies report an optimal effect of newer biologic agents on CTD-ILDs. This poses a question of how to select patients to treat with biological agents considering their underlying ILD. In this context, radiological characterization and histological classification of lung disease may be prior to helping in resolving the question [81].

## 6. Cancer-Associated Myositis

Cancer-associated myositis (CAM) is defined as malignancy that occurs within 3 years of IIM onset [82].

The first evidence in literature of this association dates back to 1916, when Stertz and colleagues published the first clinical case where the association between DM and gastric cancer tumor was suggested [83]. Since then, many studies have been conducted with the aim of clarifying this association; namely, a recent meta-analysis has tried to identify clinical factors associated with cancer risk in the idiopathic inflammatory myopathies (IIMs) [82].

The analysis of 67 articles identified 15 factors significantly associated with cancer risk: in particular, DM subtype, older age, male sex, dysphagia, cutaneous ulceration, and anti-transcriptional intermediary factor-1 gamma positivity have been associated with a greater risk of cancer, while PM and clinically amyopathic DM subtypes, Raynaud’s phenomenon, interstitial lung disease, very high serum creatine kinase or lactate dehydrogenase levels, and anti-Jo1 or anti-EJ positivity have been associated with significantly reduced risk of cancer [82]. Hill et al. evaluated 618 cases of DM and 914 cases of PM, reporting cancer rates of 32 and 15%, respectively, in their series. Non-Hodgkin’s lymphoma and ovarian, lung, pancreatic, stomach, and colorectal cancers were the most common types encountered [84]. As reported by Zahr et al., most common type of DM-related lung cancer is represented by small-cell lung cancer (29%), followed by squamous cell carcinoma (21%) and adenocarcinoma (8%) [85]. 

Neoplasms may be present in patients either before, after, or during diagnosis of myositis. Although the increased incidence of cancer in patients with DM and PM has been well established, the underlining pathogenesis still remains unknown. The most accredited theory suggests the paraneoplastic nature of DM and PM, related to an increased bioactive mediator production of an underlying occult tumor, which induces immune responses against muscle fibers and skin; this contrasts with the increased risk of cancer found in patients with idiopathic ILD, where the increased neoplastic risk is associated with the same risk factors that contribute to the genesis of IPF. To confirm this, IPF is associated with an increased risk of lung cancer rather than tumors in other sites and specifically with adenocarcinoma, which tends to develop in areas with greater fibrosis, suggesting a different pathogenesis [86].

Therefore, in light of these conclusions, we strongly emphasize the importance of routine age-appropriate cancer screening in patients with the aforementioned risk factors.

## 7. Treatment

No clear guidelines to therapy for myositis-associated ILD have been established; therefore, the treatment strategies are based on experience. In current clinical practice, the treatment of myositis-ILD consists of an induction-maintenance treatment protocol, according to ILD severity. Induction therapy involves oral or intravenous corticosteroids, followed by maintenance therapy depending on the initial response, with gradual reducing of glucocorticoids and the addition of other immunosuppressive agents (e.g., mycophenolate mofetil, azathioprine) [87]. For proper treatment, it is necessary to divide the patients according to the clinical phenotype.

In mild-moderate ILD (usually a subacute form)**,** where the interstitial disease is minimal and non-progressive, corticosteroids may be avoided. Most patients with progressive mild-moderate disease respond well to immunosuppression [62].

In progressive and/or severe ILD (subacute or acute form, mainly represented by patients having a greater respiratory deterioration), the induction therapy requires intravenous corticosteroids with either intravenous rituximab or cyclophosphamide, depending on clinical severity, comorbidities, and concomitant infection. 

RP-ILD (generally acute): these patients require intensive immunosuppressive treatment. The induction therapy consists of potent immunosuppression with a combination of immunosuppressive agents, such as high-dose glucocorticoids with combination agents (cyclophosphamide, calcineurin inhibitors and/or rituximab) [87]. Patients should be managed in an intensive care unit setting, due the high mortality rate, and with organ support when necessary, such as Extracorporeal Membrane Oxygenation (ECMO). Intravenous immunoglobulin can be used as an adjuvant therapy if there is a concurrent infection or severe myopathy; plasma exchange has been useful in some patients with anti-MDA5 disease.

Despite immunosuppression is the mainstay of treatment for patients with myositis-ILD, recent articles are gradually emphasizing the potential benefits for patients with CTD-ILD and progressive phenotype of pulmonary fibrosis. More in detail, several studies have demonstrated the efficacy of antifibrotic drugs (pirfenidone and nitedanib) in the treatment of IPF [88]; for other fibrosing non-IPF disease, several trials have been performed in the last years. Based on results of INBUILD study [89], nintedanib has been approved not only for IPF treatment, but also to treat chronic fibrosis ILDs (other than IPF) with a progressive phenotype. The exact number of patients with ILD-myositis is not known in this study; therefore, the results are difficult to extrapolate directly to patients with myositis. 

A prospective study of pirfenidone in patients with RP-ILD associated with CADM demonstrated that adding pirfenidone has no impact on the survival of acute ILD patients (disease duration < 3 months) (50% vs 50%, *p* 0.3862); whereas in sub-acute ILD patients (duration of disease 3–6 months), the pirfenidone adjunct (in 10 cases) had a significantly higher survival rate compared with the control subgroup (9 subjects) (90% vs 44.4%, *p* 0.0450) [62].

## 8. Conclusions

Idiopathic inflammatory myositis (IIM) comprises heterogeneous diseases characterized by various degrees of muscle inflammation and systemic involvement. Over the years, several autoantibodies have been identified, and their positivity determines different clinical manifestations and disease progression. ILD development leads to significant morbidity and mortality, so that early HRCT detection of lung abnormalities and recognition of risk factors for disease progression are critical to identify patients that should need medical therapy. Although the majority of patients benefit from immunosuppressive agents, there is a group of patients in which pulmonary involvement has fatal progression. In this view, further research is necessary to better evaluate the molecular profile and its association with the development and progression of lung disease, in order to achieve targeted therapies. 

## Figures and Tables

**Figure 1 medicina-58-01757-f001:**
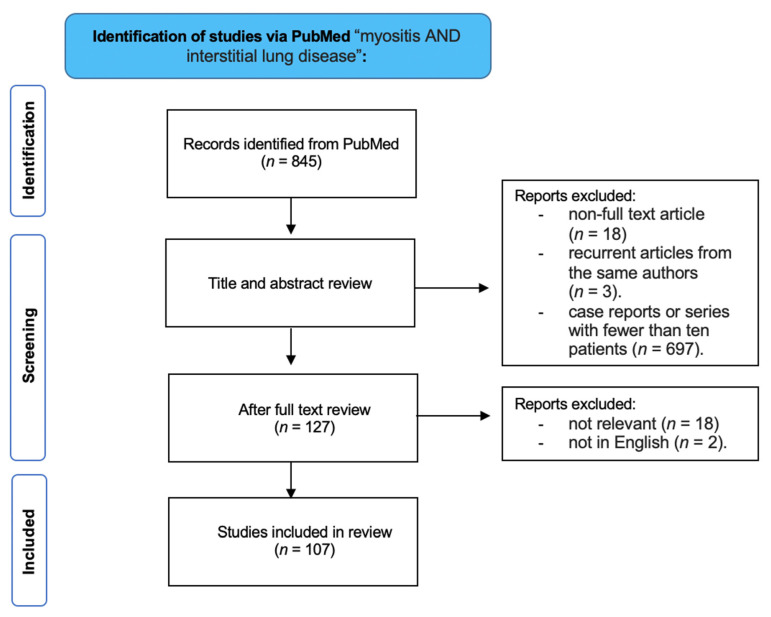
Flowchart of study selection.

**Figure 2 medicina-58-01757-f002:**
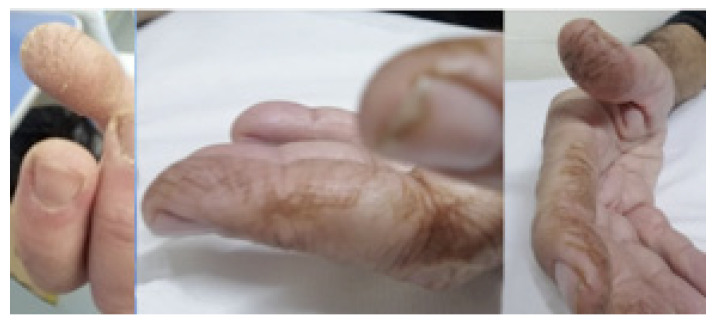
Mechanic’s hands: a dermatological findings of myositis-roughened, cracked skin at tips and lateral aspects of the fingers resulting in irregular, dirty-appearing lines.

**Figure 3 medicina-58-01757-f003:**
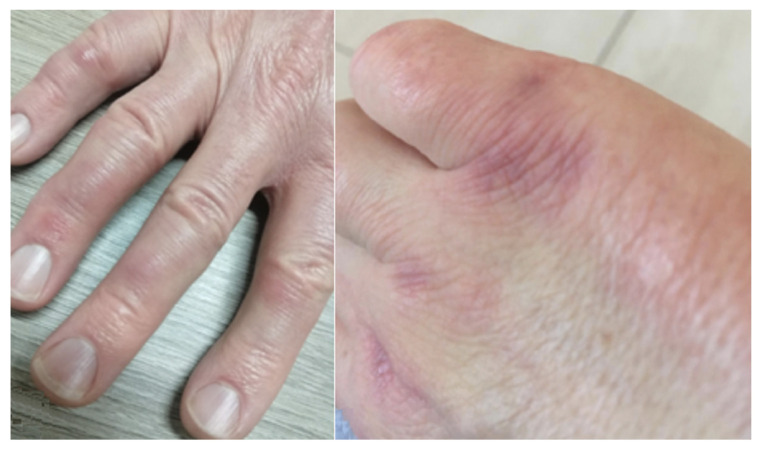
Gottron’s rash and papules.

**Figure 4 medicina-58-01757-f004:**
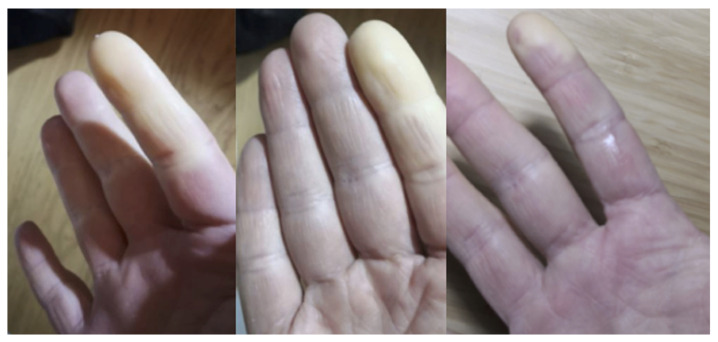
Raynaud’s phenomenon (RP).

**Figure 5 medicina-58-01757-f005:**
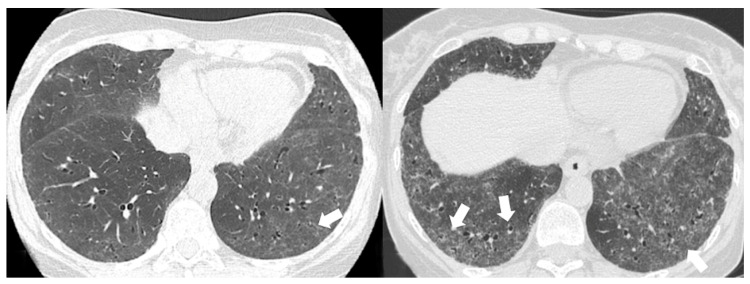
Non-specific interstitial pneumonia (NSIP) pattern: diffuse ground-glass opacities in the lower lobes and bronchiectasis (white arrow).

**Figure 6 medicina-58-01757-f006:**
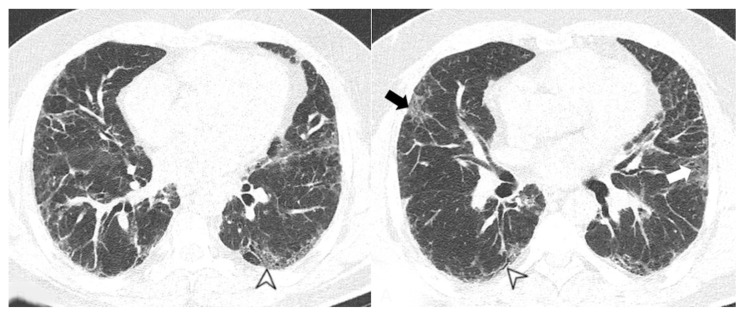
Fibrotic non-specific interstitial pneumonia (NSIP) pattern: patchy ground-glass opacities with a predominantly subpleural distribution (black arrow), fine reticulations (white arrow), and traction bronchiectasis (arrowheads).

**Figure 7 medicina-58-01757-f007:**
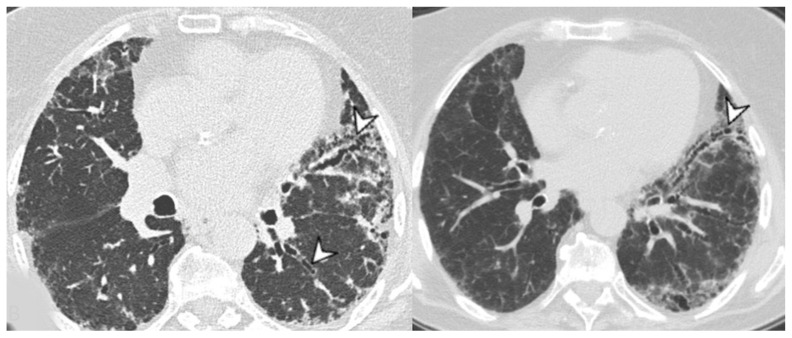
Usual interstitial pneumonia UIP pattern: traction bronchiectasis (arrowheads) and thickening of bronchovascular bundles.

**Figure 8 medicina-58-01757-f008:**
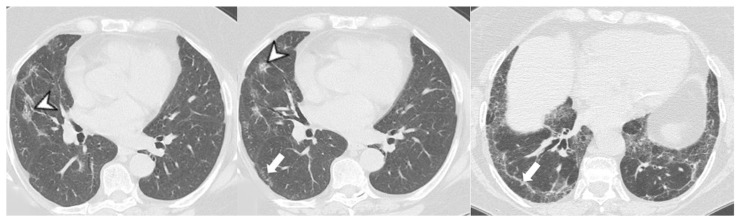
Mixed non specific interstitial pneumonia-organizing pneumonia (NSIP-OP) pattern: peripheral areas of ground-glass (arrowheads) and parenchymal bands (white arrows).

**Figure 9 medicina-58-01757-f009:**
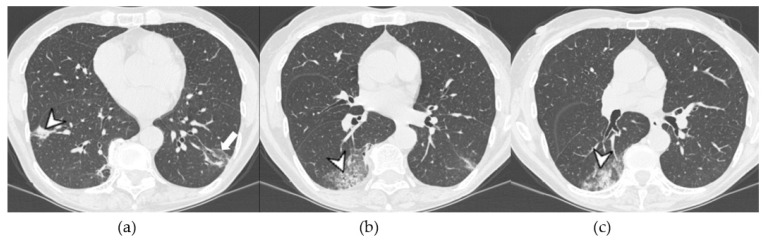
OP pattern: (**a**) patchy consolidation (arrowhead), (**b**) ground-glass opacities (arrowhead), and (**c**) consolidation (arrowhead) and initial reversed halo sign (white arrow).

**Figure 10 medicina-58-01757-f010:**
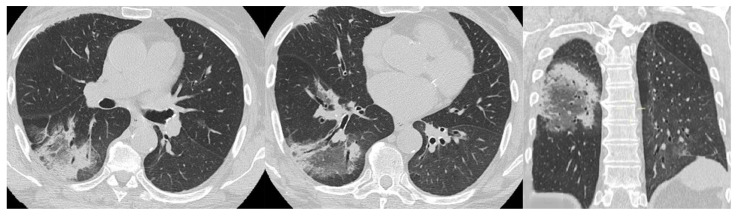
OP pattern with reversed halo sign.

**Figure 11 medicina-58-01757-f011:**
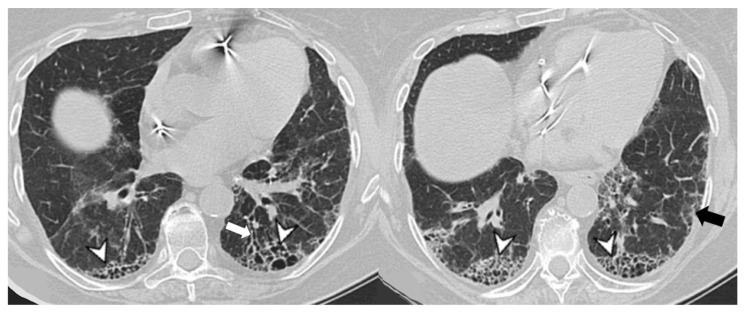
UIP pattern: honeycombing (arrowhead), traction bronchiectasis and bronchiolectasies (white arrow), and pleural reticulation (black arrow).

**Figure 12 medicina-58-01757-f012:**
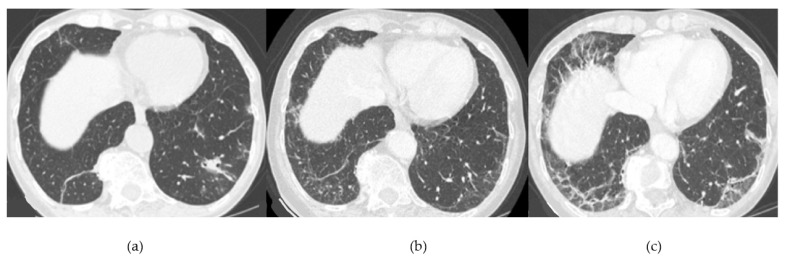
Progressive ILD: progressive pulmonary involvement with reproducing NSIP-OP pattern. (**a**) HRCT at onset disease shows irregular linear opacities, subpleural reticulation, and consolidation. (**b**) HRCT taken 12 months later shows increase of subpleural reticulations. (**c**) HRCT taken 24 months later shows further increase of the interstitial involvement with ground-glass opacities.

**Figure 13 medicina-58-01757-f013:**
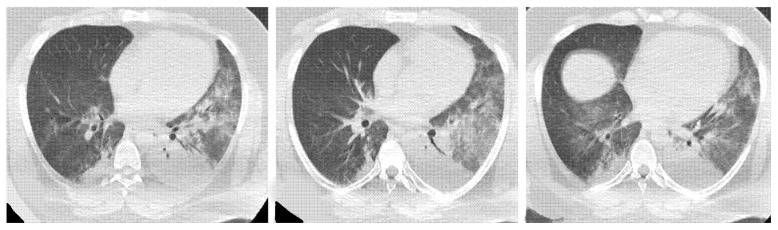
Aspiration pneumonia: patchy ground-glass and bilateral dorsal consolidation with positive air bronchograms.

**Figure 14 medicina-58-01757-f014:**
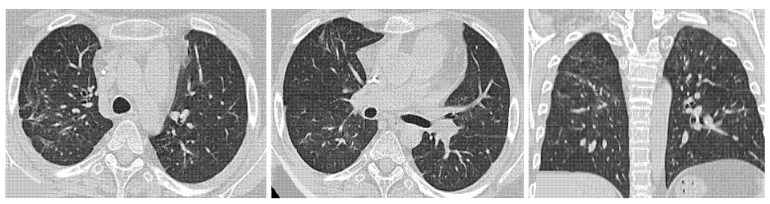
ILD-cyclophosphamide related presentation: diffuse ground-glass opacities in the early stage especially represented in the upper lobes.

**Table 1 medicina-58-01757-t001:** Summary of laboratory data and radiological features for patients with PM-DM.

PM-DM	
**Laboratory findings**	Muscular enzymes: creatine kinase (CK) aldolase myoglobin lactic dehydrogenase aspartate transaminase Autoantibodies: Myositis-Specific: Jo1, PL7, PL12, EJ, OJ, ZO, HA Myositis-Associated: Anti-Ro52, Pm/Scl, U1-RNP, anti-Ku
**Clinical features**	Muscle weakness: acute or subacute onset symmetric, diffuse, predominantly proximal Arthritis: asymmetric and oligoarticular
	Cutaneous manifestation: DM: Gottron’s rash and papules, Shawl Sign, Holster Sign and V Sign, mechanic’s hands, hicker’s feet, heliotrope rash Subcutaneous calcinosis (especially in juvenile forms) Raynaud’s phenomenon (RP)
	Lung involvement: rapidly progressive dyspnea and respiratory failure; insidious onset of dyspnea hypoventilation and respiratory failure due to respiratory muscle involvement; aspiration pneumonia;
**Radiological features**	ILD: NSIP UIP OP pattern DAD NSIP-OP

**Table 2 medicina-58-01757-t002:** List of the clinical course, lesions, distribution, and HRCT pattern across myositis subtypes.

		Anti-ARS-Abs	Anti-MDA-5 Abs
Clinical course		subacute to chronic	acute rapidly progressive to subacute
Lesions		GGO, reticulations, consolidations	consolidations, GGOs
Distribution		Homogeneous; lower lung lobes, along bronchovascular bundles and lung periphery; loss of volume of lower lobes	Patchy; peripheral lower lobes or along the bronchovascular bundles
CT pattern	NSIP OP NSIP-OP UIP DAD-unclassifiable	50% 20% 25% 10% +/−	20% 50% 25% <5% ++
Prognosis		good response to treatment; possibility of relapses	Poorer prognosis

## Data Availability

Not applicable.

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
