# Peer review of "Clinical and Radiological Features of Interstitial Lung Diseases Associated with Polymyositis and Dermatomyositis"

_medicina, 2022, doi:10.3390/medicina58121757_

Round 1

Reviewer 1 Report

The topic of this paper is very actual and has high interest.

Even the title of the paper is "Clinical and radiological features of interstitial lung diseases", the introduction has only 1 sentence about the imaging features of the ILDs. Please add more to the last section of the inroduction about the imaging modalities.

The abbreviations are not resolved, eg. NSIP, OP etc.

Please add the role of chest XRay to the first sentence of the imaging features.

On the Figure 5., there is a fibrotic NSIP pattern, not only NSIP.

Figure 6 looks like not (fibrotic?) NSIP pattern, it seems as pUIP, was it histiologically proven?

Please add the meaning of HRCT to the Radiological features section. Please add the role of the PCCT (photon counting CT) as well.

The importance of pulmonary hypertension is missing, please add it to the text (eg. Clin Rheumatol. 2015 Dec;34(12):2105-12.)

Lung cancer is frequently developing in IPF, please add this information to the manuscript (eg. Clin Resp J 2018 Apr;12(4):1700-1705.)

Please add the therapy of ILD as last paragraph of the paper too (eg. Eur Respir Rev. 2019 Sep 4;28(153):190021.)

The occurance rate of ILD patterns are missing in the Table 1., please add the % rate to the table.

Author Response

Dear Reviewer,

you find attached our revisions, based on your precious suggestions.

Hoping that everything request will be satisfied,

Best regards

Reviewer 2 Report

Review report for the article Clinical and radiological features of interstitial lung diseases 2 associated with polymyositis and dermatomyositis

Manuscript Ref No medicina-2003675

1. Recommendation

Major revision

2. Overview and general comments

The manuscript aims to review clinical and radiological features of interstitial lung disease (ILD) in dermatomyositis and polymyositis. It is good to take such an initiative because inflammatory myopathies (IM)/myositis are with scleroderma the commonest rheumatic diseases manifesting with ILD and ILD is a major cause of mortality in myositis patients. However, the manuscript lacks originality and methods as well as citations included are not so credible(because only 2021 and 2022 there aremany expert reviews on ILD in IM althoough they mostly addressed management issues, for instance, i do not see these ones in the manuscript: doi: 10.3390/medicina57040347 and 10.1016/j.berh.2022.101769). This leads me to recommend a major revision, so that your review can really be original and used by myositis-ILD experts.

3. Specific comments

3.1. Major comments

-Introduction: please, decrease its lenght to max three short paragraphs. Focus on ILD in myositis (what is known and what is unknown), and then state the review objective and move on to the body of the review.

-Body of the review: please, start by a small methods section (describing the search strategy used, periods and it would be great if you could add a figure describing the study selection process). Then, move on to the topic with clinical features as the first one, and radiological features as the second one. When describing clinical features, please, forget about extra-pulmonary manifestations not linked to ILD (including autoantibodies not  linked to ILD) or give them a very little weight in the text. So, figures of Raynaud's phenomenon and others are not really relevant here. I was surprised you did not emphasize on pulmonary manifestations such as dyspnea on exertion, bibasal lung crackles and on extra-pulmonary manifestations such as digital clubbing/hypertrophic osteoarthropathy which is highly prevalent in non-rheumatic ILD and for which little is known regarding rheumatic diseases: is it also associated with the prognosis? See Essouma et al. See J Clin Rheumatol.2021. I wondered why you focused on dermatomyositis and polymyositis. What are about other myositis subtypes (e.g. anti-synthetase syndrome which with DM are the commonest IM subtypes with ILD). Even among DM subgroups, I did not really see how you emphasize on MDA5-positive Dm-associated ILD which is the most severe and fatal form of myositis-associated ILD and which likely shares many features with COVID-19 (See Latika et al. Rheumatol Int. 2021) and macrophage associated syndrome. We would like to see how ILD differs in presentation across myositis subtypes (in a table) and radiological images. It is treu that all ILD patterns on radiology can be seen in myositis. But, which one is the most important in each subtype? Please, do not mix pneumonia with ILD. You could rather add a paragraph on pulmonary comorbities of ILD in myositis patients. Very important, please make comments about how ILD affects the prognosis of myositis patients. I also wondered I did not see the difference commonly made in the literature between chronic and rapidly progressive ILD (which myositis subtypes are affected?) Is there a different view of ILD between rheumatologists/pneumologists/clinicians interested in myositis-ILD and radiologists (This would be interesting because I see that the special issue is led by a radiologist)?

3.2. Minor comments

-Use this link for example, to update your references: https://pubmed.ncbi.nlm.nih.gov/?term=myositis+AND+interstitial+lung+disease&filter=pubt.meta-analysis&filter=pubt.review&filter=pubt.systematicreview&filter=years.2021-2022 and use expert references mostly, to give chances to your articles to be useful in the field.

-The abstract will need update as well.

                                                                         Anonymous report

                                                                         Available online on 1 Nov 2022

Author Response

(The authors gave the same response as above.)
